

# Organic soil amendments using vermicomposts under inoculation of N$_2$-fixing bacteria for sustainable rice production

Mehdi Ghadimi[1], Alireza Sirousmehr[1], Mohammad Hossein Ansari[2] and Ahmad Ghanbari[1]

[1] Department of Agronomy, University of Zabol, Zabol, Sistan-o-Baluchestan, Iran
[2] Department of Agronomy, Rasht Branch, Islamic Azad University, Rasht, Iran

## ABSTRACT

Organic and biological fertilizers are considered as a very important source of plant nutrients. A field experiment was conducted during 2017−2018 in paddy soil to investigate the effect of vermicomposting of cattle manure mixture with *Azolla* and rice straw on soil microbial activity, nutrient uptake, and grain yield under inoculation of N$_2$-fixing bacteria. Experimental factors consisted of organic amendments at six levels (vermicomposts prepared from manure (VM); manure + rice straw (VRM); manure + *Azolla* mixture (VAM); manure + rice straw + *Azolla* mixture (VRAM); raw manure without vermicomposting (M), and a control) and N$_2$-fixing bacteria at three levels (*Azotobacter chroococcum*, *Azospirillum brasilence*, and non−inoculation). The results showed that, vermicompost treatments compared to control and raw manure significantly increased the number and biomass−C of soil microorganisms, urease activity, number of tillers hill$^{-1}$, phosphorus (P) and potassium (K) uptake, and grain and protein yield. Inoculation of plants with N$_2$-fixing bacteria, especially *Azotobacter* increased the efficiency of organic amendments, so that the maximum urease activity, soil microbial activity, P and N uptake, and grain yield (4,667 (2017) and 5,081 (2018) kg/h) were observed in vermicompost treatments containing *Azolla* (VAM and VRAM) under inoculation with *Azotobacter*. The results of the study suggested that, using an organic source along with inoculation with appropriate N$_2$-fixing bacteria for vermicompost has a great effect on enzyme activity, soil biology, nutrient uptake and grain yield has a synergistic interaction on agronomic traits under flooded conditions. Therefore, this nutrient method can be used as one of the nutrient management strategies in the sustainable rice production.

## INTRODUCTION

The use of organic and biologic fertilizers in rice production is one of the most important methods used for soil fertility in a sustainable system (*Wang et al., 2015*; *Amanullah, 2016*; *Chen et al., 2018*). Among various organic fertilizers, vermicompost superiority

Corresponding author
Alireza Sirousmehr,
asirousmehr@uoz.ac.ir

has been proved in maintaining the long-term soil fertility (*Lim et al., 2012*; *El-Haddad et al., 2014*; *Guridi et al., 2017*; *Sharma & Garg, 2018*). Actually, soil amendment with vermicompost is an agronomically interesting practice as well as an attractive waste management strategy (*Ludibeth, Marina & Vicenta, 2012*; *Baghbani-Arani & Modarres-Sanavy, 2017*). Production of vermicompost as an easy and nature-friendly technology is a semi-aerobic process carried out by a specific group of earthworms (*Eisenia fetida*) and some soil microorganisms, especially bacteria and actinomists used for production of organic fertilizers from the waste and stabilization of these materials (*Mahanta et al., 2012*; *Lim, Lee & Wu, 2016*). As a result, stabilization reduces the environmental problems associated with manure (such as salinity) by transforming it into safer and more stabilized material for soil amendment (*Mahanta et al., 2012*). In addition to less time for bioconversion, the presence of higher content of minerals, hormones, enzymes, humic acids and bioavailable form of nutrients has been reported for vermicompost (*Campitelli, Velasco & Ceppi, 2012*; *El-Haddad et al., 2014*; *Khan, 2018*). Vermicompost contains nutrients in such forms that are readily available to the crops, such as nitrates, exchangeable P, soluble K, iron (Fe), calcium (Ca), magnesium (Mg), etc. (*Tejada & González, 2009*; *Ludibeth, Marina & Vicenta, 2012*). *Guridi et al. (2017)* reported that, vermicomposts contain biologically active substances such as plant growth regulators, and have great potential in maintaining the soil fertility. If vermicomposts are integrated in nutrient management in agricultural fields, the costs of crop production may be reduced significantly (*Lim, Lee & Wu, 2016*; *Sharma & Garg, 2018*). On the other hand, some composts and vermicomposts (especially those made from chaff of wheat or rice straw straw)with high value of carbon –to- nitrogen (C/N) may increase the time of bioconversion of the crop or stimulate N fertilization in the soil, and face the plant with N deficiency in some growth stages (*Zhu et al., 2013*; *Li et al., 2015*; *Eo & Park, 2019*).

The use of *Azolla pinnata* (a free-floating weed) is a solution for increasing the N composts and vermicomposts. In terms of nutrients, the amount of *Azolla* nutrient has been shown to vary in different periods of time and has an average of 5.3% N, 8.3% K and 0.6% Mg, and is free from lead, mercury or arsenic (*Sreenivasa, 2012*). It has been reported that, vermicomposting of the straw is mixed with *Azolla* along with cattle dung to obtain a better quality crop suitable for agricultural applications (*Arora & Kaur, 2019*).

The use of bacteria with the capability for Biological N Fixation (BNF), such as *Azospirillum*, *Herbaspirillum*, and *Azotobacter* , as an effective strategy is another solution for increasing N amount in paddy soil, under nutrient management for production of safer and cheaper rice (*Ladha & Reddy, 2003*; *Sanati et al., 2011*; *Zhang et al., 2017*). In this regard, *Bgattacharjee, Singh & Mukhopadhyay (2008)* showed an increase in host-plant N content, up to 30–45 mg of N plant$^{-1}$ (6-week-old seedlings) resulted from N$_2$-fixing. This group of bacteria, in addition to the N$_2$-fixing capability in a cooperative way, contributes in solving the nutrients such as P, K ,and iron (Fe), and also has the capability for producing Phytohomones, Vitamins and Siderophores (*Zhang et al., 2018*). Due to their positive effects on plant growth stimulation, these microorganisms are called as Plant Growth Promoting Rhizobacteria (PGPR) (*Wu et al., 2005*). *Mahanta et al. (2012)* revealed the superior performance of PGPR in increasing rice growth and grain yield and

**Table 1  Results of physio-chemical analysis of the experimental soil.**

| Year | Sand | Silt | Clay | EC | pH | Organic Carbon | N | P | K | Zn | Fe | Mn |
|------|------|------|------|------|------|------|------|------|------|------|------|------|
| | (%) | | | (dS/m) | | (%) | (%) | | | (mg/kg) | | |
| 2017 | 29.8 | 26.3 | 43.9 | 1.05 | 7.6 | 0.75 | 0.71 | 15.8 | 164 | 0.89 | 4.41 | 6.21 |
| 2018 | 28.5 | 32.2 | 39.3 | 1.24 | 7.1 | 0.86 | 0.66 | 12.4 | 194 | 1.23 | 5.82 | 7.51 |

improving soil health in addition to saving 40–80 kg N ha$^{-1}$. In this regard, there is great interest in exploring the diversity of PGPR as substitutes for some chemical agricultural inputs (*Rodrigues, Ladeira & Arrobas, 2018*). Generally, the N-transformation processes in the rice rhizosphere include N mineralization, denitrification, N fixation, and ammonia volatilization. Microbial-mediated mineralization and BNF are very important to the level of available N content in the soil and N uptake by rice (*Pattnaik et al., 1999*; *Nayak, Babu & Adhya, 2007*; *Kumar, Swain & Bhadoria, 2018*; *Zhang et al., 2018*), so an increase in the abundance of microbial populations could result in a faster N mineralization rate and increased N availability for plants (*Li et al., 2015*; *Chen et al., 2017*; *Khan, 2018*).

In Iran, rice is the second source for supplying food after wheat, with an annual consumption of 39.4 kg per person (*Ashoori et al., 2018*). Since Iran is located in a warm and dry region, its soils have low organic matter, and it is noteworthy that, the organic matter of the soil as the most important factor in crop yield ranks after the water . On the other hand, 20 million tons of manure is produced by animals per annum in Iran. The rate of compost and vermicompost production is negligible compared to manure. Efforts have been taken in recent years to produce more vermicompost, but its application in farms, especially in rice farms has not been increased remarkably (*Rezaei, 2013*; *Taheri Rahimabadi, Ansari & Razavi Nematollahi, 2018*). In the present study, cow manure vermicomposting process was done in different mixtures with *Azolla* and rice straw under inoculation of N$_2$-fixing bacteria aimed at improving the agronomic value of the vermicomposts on promoting biological activity of paddy soil, nutrient uptake and grain yield of rice.

# MATERIALS AND METHODS

## Experimental site and plant growth conditions

The field experiment was conducted on clay-loam soil at the Rice Research Institute Farm of Rasht, Guilan province, Iran during 2017–2018. The area is located at 37°22N latitude and 49°63E longitude and 15 m above the sea level. To simplify the comparison of the growing season weather, the monthly total precipitation and temperature were considered from May to September at the Rasht Agricultural Research Farm (Figs. 1A and 1B). To determine soil characteristics, soil sampling was performed before the experiment. To do this, field soil sampling was done from the depth of 0–30 cm in 8 spots. Then, the collected samples were sent to the laboratory to determine soil texture and chemical composition. Properties of experimental soil samples are given in Table 1.

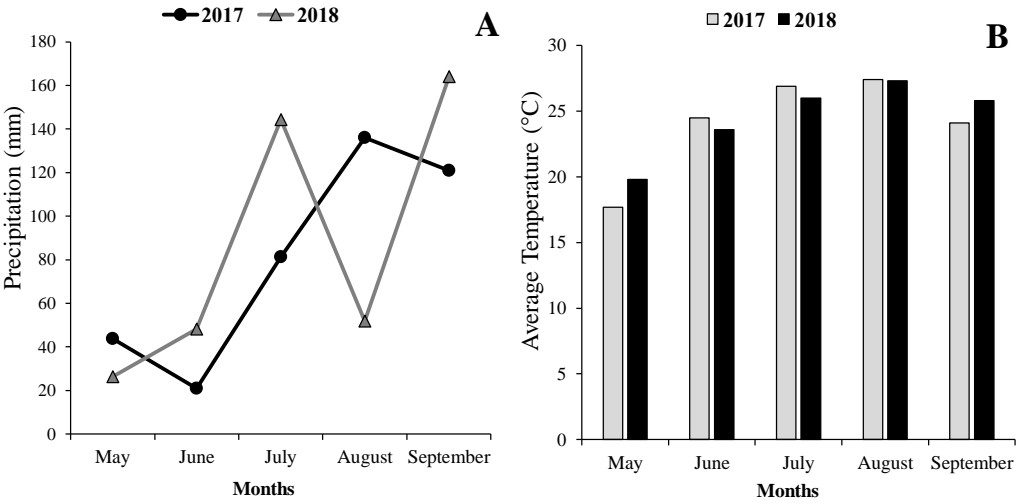

**Figure 1** Monthly rainfall (A) and temperature (B) from May to September (2017–2018) at the Rasht, Iran.

**Table 2** Organic amendments studied in present experiment.

| | |
|---|---|
| VM | Vermicompost produced from manure alone |
| VRM | Vermicompost produced from mixed of manure and rice straw[*] |
| VAM | Vermicompost produced from mixed of manure and *Azolla* |
| VRAM | Vermicompost produced from mixed of manure, rice straw and *Azolla* |
| M | Raw manure without vermicomposting |

**Note.**
*The volume of rice straw and Azolla used to produce different vermicomposts was one tenth of the weight of manure.

## Experimental design and treatments

The experiments were carried out in a factorial trial based on Randomized Complete Block Design (RCBD) with three replications. Experimental factors consisted of organic amendments at six levels (M, VM, VRM, VAM, VRAM 10 t/hm$^2$, and a control) and N$_2$-fixing bacteria at three levels (*Azotobacter chroococcum*, *Azospirillum brasilence*, and non-inoculation). The organic amendments that included of manure and vermicomposts prepared from manure alone or mixed with different materials are presented in Table 2.

## Preparation of vermicompost

In the present study, rice straw was collected from paddy fields and dried and chopped prior to use in the experiment. Urea fertilizer consisting of urea was prepared from livestock farms near the test site. *Azolla* was just harvested after rice was harvested from rice fields.

A tank (3 m$^3$ in size) was used to produce vermicompost, which contained a population of active and stable earthworms (*Eisenia andrei*). Fertilizer application was performed on a reactor containing active earthworms. A mixture of cattle manure and rice straw was poured into the reactor, which supported a population density of 250 g of earthworm's kg$^{-1}$ in the top layers. The upper surface of the reactor was divided into four chambers,

**Table 3  Chemical analysis of the cattle manure and its vermicomposts.**

| Parameter[*] | M | VM | VRM | VAM | VRAM |
|---|---|---|---|---|---|
| pH | $8.10 \pm 0.13$ | $7.43 \pm 0.06$ | $7.60 \pm 0.00$ | $6.83 \pm 0.03$ | $7.7 \pm 0.01$ |
| EC | $1.08 \pm 0.01$ | $1.26 \pm 0.03$ | $0.96 \pm 0.01$ | $1.44 \pm 0.01$ | $1.38 \pm 0.04$ |
| Ash content (g/kg) | $197 \pm 10.3$ | $441 \pm 21.7$ | $239 \pm 7.09$ | $311 \pm 14.28$ | $291 \pm 8.20$ |
| Total OC (g/kg) | $412 \pm 16.9$ | $316 \pm 6.5$ | $439 \pm 16.9$ | $293 \pm 8.92$ | $336 \pm 9.03$ |
| Total OM (%) | $74.6 \pm 2.3$ | $49 \pm 4.3$ | $87.6 \pm 2.3$ | $45.7 \pm 3.40$ | $46.6 \pm 1.12$ |
| Total N (g/kg) | $16.8 \pm 0.51$ | $25.3 \pm 1.54$ | $21.6 \pm 0.51$ | $32.4 \pm 1.55$ | $27.2 \pm 2.03$ |
| Total P (g/kg) | $9.4 \pm 0.91$ | $15.4 \pm 0.84$ | $12.7 \pm 0.91$ | $17.2 \pm 0.76$ | $11.82 \pm 0.07$ |
| Total K (g/kg) | $10.4 \pm 1.02$ | $13.7 \pm 1.11$ | $10.1 \pm 1.02$ | $14.8 \pm 0.34$ | $13.46 \pm 0.73$ |
| C:N | $24.46 \pm 2.3$ | $11.3 \pm 0.47$ | $17.14 \pm 1.06$ | $9.31 \pm 0.24$ | $12.35 \pm 1.34$ |
| Cu (mg/kg) | $135.2 \pm 7.92$ | $164.8 \pm 13.8$ | $152.0 \pm 17.06$ | $169.7 \pm 11.09$ | $155 \pm 7.66$ |
| Fe (mg/kg) | $215.4 \pm 5.8$ | $416.6 \pm 5.7$ | $372.1 \pm 21.3$ | $542.3 \pm 13.76$ | $484 \pm 9.23$ |
| Mn (mg/kg) | $109.7 \pm 11.4$ | $248.7 \pm 4.6$ | $212.0 \pm 6.9$ | $307.6 \pm 9.51$ | $260 \pm 8.59$ |
| Zn (mg/kg) | $184 \pm 9.83$ | $369 \pm 12.06$ | $342 \pm 5.43$ | $289.7 \pm 14.5$ | $319 \pm 14.25$ |

**Note.**

[*]Dry matter basis. Data are presented as mean $\pm$ standard error (n = 3)

and 45 kg of cattle manure along with rice straw were placed in three layers on top of each other (15 kg per layer) in each chamber, which was processed by earthworms. The moisture content of cattle manure in the vermi-reactor was maintained in the range of 75 to 80% and the sample was harvested from the last layer (two months after tillage) of the reactor (*Lazcano, Gómez-Brandón & Domínguez, 2008*).

The Vermicomposting process caused a significant change in cattle manure. The vermicompost was much darker in color and well-formed, and after the earthworm's activity was processed into a homogeneous compound. The vermicomposting process helps to significantly reduce the pollution of organic matter in the environment and soil. Table 3 presents the physical, chemical, and nutritional properties of cattle manure and vermicompost. Cattle manure and vermicomposts were applied to the specified treatments 20 days before transplanting and mixed thoroughly with the soil. In the control, according to the results of soil decomposition, the required elements of chemical fertilizers were provided.

## Bacterial inoculant preparation and sowing conditions

*Azospirillum brasilense* (strain BBU168) and *Azetobacter chroococcum* (strain SUAA4) isolated from rice roots in the paddy soils have been developed by the Soil Microbiology Research Institute of Soil and Water Research. The bacteria were cultured in improved medium and purified to a final concentration of $10^8$ CFU ml$^{-1}$. In the present study, the cultivated rice cultivar was Hashemi, the growth period of which varies from 118 to 140 days from planting date to physiological maturity. In order to treat the plants with bacteria, the roots of rice seedlings (twenty-seven days old) were placed separately in solutions containing 1000 ml of suspension of bacteria and bacteria for 2 h. In addition, for plants without inoculation, the roots of rice seedlings were placed in saline solution. Immediately after applying the bacterial treatments, the seedlings were planted at $0.2 \times 0.2$

m (three seedlings per mound) on the eighth of May. Weeds were controlled manually during the growing period and no herbicides were used. .During the rice-growing period, the floodplain water level of the plateaus was constantly adjusted to 10 cm.

## Soil bacterial account and microbial biomass-C

The soil of the rice rhizosphere was sampled at the tillering stage. Rhizosphere soil samples were collected by inserting a cylindrical cylinder 10 cm in diameter with a sharp edge at the bottom.

Soil microbial biomass-C was measured by modified chloroform fumigation–extraction method with fumigation at atmospheric pressure (*Witt et al., 2000*).. Soil samples, 35 g on an oven-dry basis (48 h at 105 C$^{\circ}$) were weighed into 500 -ml glass Schott bottles and were fumigated by adding 2 ml of ethanol-free chloroform directly onto the soil. Microbial biomass C was estimated by extracting the fumigated soil with 0.5 M $K_2SO_4$ and extractable C determined by modified dichromate digestion of soil extract (*Vance, Brookes & Jenkinson, 1987*).

The total number of bacteria was determined using plate cultures and direct counting of microbes (*Kelly, Haggblom & Tate, 1999*). 2 10 -g samples from each plot were sprayed separately into 250 -ml Erlenmeyer containing 95 ml of sterile distilled water. The Erlenmeyer was rotationally mixed for 5 min, then they were fixed for 15 s allowing coarse particles to be deposited. 5 experimental tubes containing 9 ml of sterile distilled water were prepared. 1 ml of soil suspension was taken and transferred to tube No. 1. Once again, 1 ml was taken from the first tube and transferred to the second tube. The next dilutions were also prepared in the same way. The soil dilution was done to a dilution of $10^{-6}$ g/ml. For bacterial counting, dilutions of $10^{-4}$, $10^{-4}$ and $10^{-4}$g/ml were selected and cultured in a special culture medium of aerobic heterotrophic bacteria, namely, Nutrient Agar with a concentration of 16 g/L. To prevent the growth of fungi, 50 mg/l of Nystatin was added to the culture medium of bacteria. After 25 h of incubation at 25 °C, colony counting was done.

## Urease activity

Urease (EC 3.5.1.5) activity was measured in growing stages of rice (5 days after transplantation (T1), $\frac{1}{4}$ maximum tillering stage (30 days after transplantation) (T2), $\frac{1}{4}$ panicle initiation stage (70 days after transplantation) (T3) and $\frac{1}{4}$ maturity (100 days after transplantation) (T4)), according to the method proposed by *Pattnaik et al. (1999)*. 20 g of air-dried soil (passed through a 2-mm sieve) was mixed with urea (20 ml) to provide a final concentration of 2,000 $\mu$g/g soil, and the suspensions were incubated for 5 h. The amount of residual urea present in the soil suspension upon incubation was determined by the non-buffer method introduced by *Zantua & Bremner (1977)*. Urease activity was expressed as milligrams of urea hydrolyzed per gram of dry soil per hour.

## Plant sampling

In the dough stage, 8 randomly chosen plants were removed from each plot, and chlorophyll was determined by Arnons' method (*1949*) in the flag leaves.
After maturity stage, plant height, panicle height, the No. tillers $\text{hill}^{-1}$ and the yield of the seed (taking into account 14% moisture) were determined from 2 square meters per plot. The moisture content of the grains was measured using digital grain moisture meter (Model GMK–303R5–Korea) and the following equation was used to calculate the grain yield per plot by considering the moisture content of 14%:

$$\text{Grain yield} = \frac{(100 - \text{moisture content of the sample}) \times \text{fresh grain weight}}{86}$$

To determine the biomass, plants 1 square meter from the center of each plot were randomly harvested and then dried and weighed separately in a paper bag at 45 °C for 48 h (reported as biological function). The harvest index was also calculated using the following equation:

$$\text{Harvest Index}(\%) = \frac{\text{Grain yield}}{\text{Biological yield}}$$

The concentration of nitrogen (N) in the grain and straw samples was obtained after digestion in acid in the digestion block by the Kjeldal method. The protein concentration in the grain and straw was also calculated from the product of the amount of nitrogen in the grain and straw at 6.25. The concentrations of P and K in the grain and straw after digestion in a 9:1 ratio ($HNO_3$:$HClO_4$) of di-acide mixture were estimated using standard methods described by AOAC (1970). P and K uptake and grain protein yield were obtained by multiplying the dry weight of the grain by their concentration.

It should be noted that all treatments applied to rice, operations performed as well as measured traits in the first year and in the second year are exactly the same.

## Statistical analysis

All data were subjected to Analysis of Variance (ANOVA) according to the methods described for factorial trial based on randomized complete block design combined over the years using SAS software 9.3. When F test indicated statistical significance at $P < 0.01$ or $P < 0.05$, the Least Significant Difference (LSD) was used to separate the means.

## RESULTS

### Urease enzyme activity

Urease enzyme activity of soil increased up to stage T3 in most organic amendments in both years of the experiment, but then (in stage T4) its activity decreased, and under non-inoculation conditions (Figs. 2A and 2B), the urease activity trend lines of the control treatment collided with the urease activity trend lines of organic amendments while, such a collision was not seen under inoculation of *Azotobacter* (Figs. 2C and 2D) and *Azospirillum* (Figs. 2E and 2F). Therefore, the effectiveness of organic amendments on enzyme activity in rhizosphere of inoculated plants increased compared to non-inoculated plants. In both years, VAM showed the most urease activity and *Azotobacter* seemed to show more enzyme activity compared to *Azospirillum* in response to the application of organic amendments.

### Soil bacterial account and microbial biomass–C

The interaction effect of bacteria × organic amendments × year on microbial carbon biomass was significant ($P < 0.01$) (Table 4). The mean comparison showed that, in the
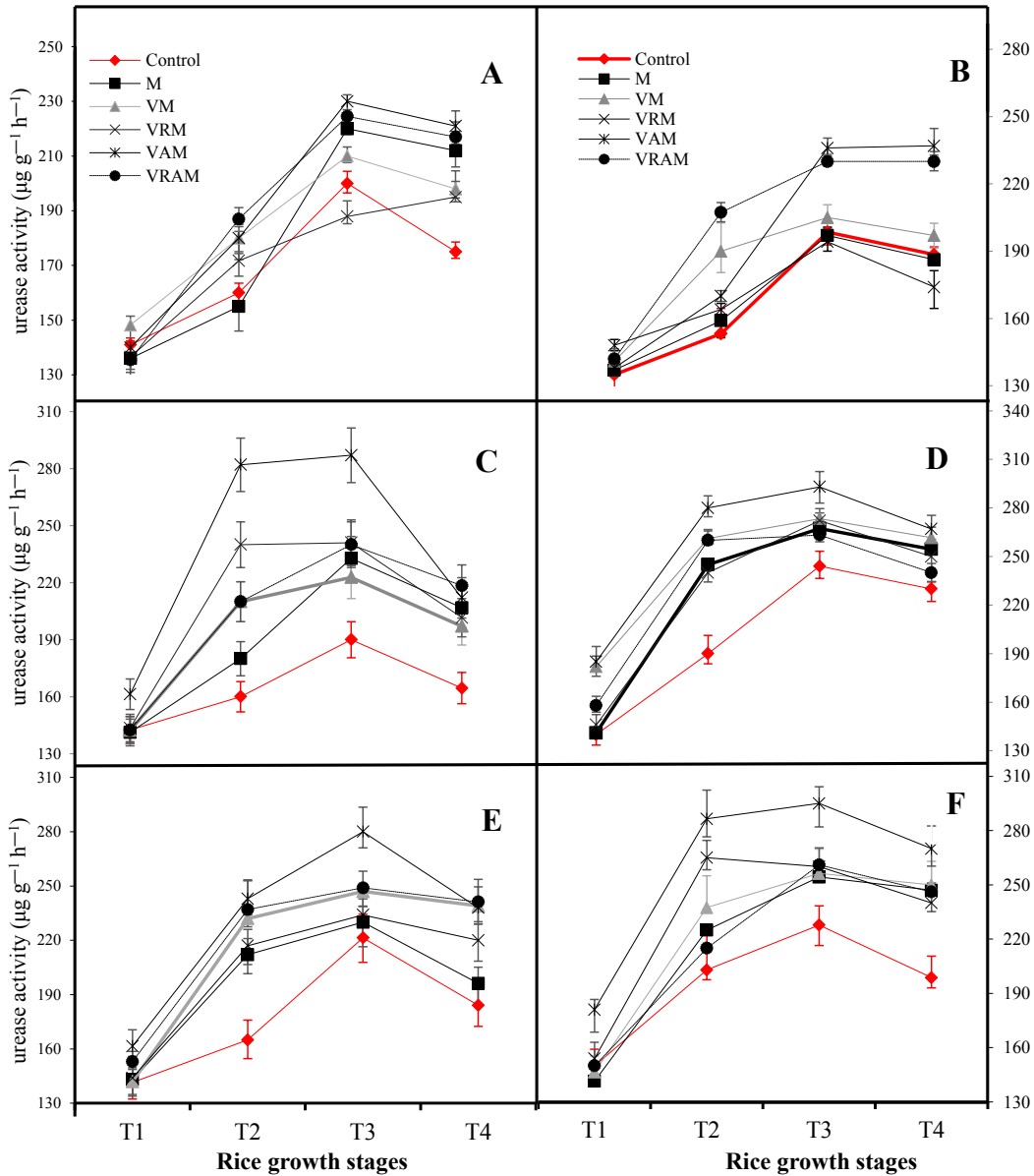

**Figure 2 Dynamics of Urease activity in a flooded rice at different growth stages of rice planted under organic amendments and N₂-fixing bacteria.** Growth stages: T1: (5 days after transplantation), T2: 1/4 maximum tillering stage (30 days after A (2017) and B (2018): non–inoculum; C (2017) and D (2018): *Azotobacter* inoculum; E (2017) and F (2018): *Azospirillum* inoculum. transplantation), T3: 1/4 panicle initiation stage (70 days after transplantation) and T4: 1/4 maturity (100 days after transplantation).

first year (2017), compared to control, organic amendments increased microbial carbon biomass (24–45%) in non-inoculated plants, plants inoculated with *Azotobacter* (42–78%) and *Azospirillum* (31–64%). While in the second year (2018), this increase was obtained as 29–41%, 68–82%, and 63–71%, respectively in non-inoculated plants, plants inoculated with *Azotobacter* and *Azospirillum* compared to control. In both years, the highest microbial

**Table 4  ANOVA of the effect of organic amendment on soil bacteria count (SBC), microbial biomass-C (MBC), chlorophyll a (CA), chlorophyll b (CB), total chlorophyll (TC), No. of tiller hill$^{-1}$ (NTH), seed yield (SY), biological yield (BY), protein yield (PY), P uptake (PU), and K uptake (KU) under inoculation with PGPRs.**

| S.O.V | Mean square (MS) | | | | | | | | | | |
|---|---|---|---|---|---|---|---|---|---|---|---|
|  | SBC | MBC | CA | CB | TC | NTH | SY | BY | PY | PU | KU |
| Y | 14934ns | 789* | 3.4* | 1.07** | 0.65ns | 0.31ns | 1595781** | 3771791** | 5698** | 35.9ns | 1108** |
| Y(R) | 12449ns | 160ns | 20.8** | 0.24** | 16.8** | 0.86ns | 3232ns | 474140ns | 790ns | 50.6ns | 93.9ns |
| B | 81900** | 73401** | 63.8** | 8.14** | 116** | 6.41* | 4341781** | 19679459** | 236472** | 660** | 818** |
| M | 30723** | 26830** | 12.0** | 2.88** | 25.3** | 136** | 5293349** | 15884979** | 37202** | 790** | 436.9* |
| Y ×B | 1892ns | 1124** | 0.09ns | 0.05ns | 0.01ns | 1.71ns | 77762** | 3324497** | 25543ns | 287* | 484* |
| Y × M | 5232ns | 232ns | 0.01ns | 0.01ns | 0.004ns | 3.23ns | 348570** | 926332** | 20414* | 114ns | 265ns |
| B × M | 23592** | 1111** | 3.13** | 1.36** | 7.74** | 18.5** | 127978** | 645059** | 57888** | 364** | 795** |
| Y × B × M | 8293ns | 614** | 0.01ns | 0.008ns | 0.40ns | 5.54* | 20420* | 910690** | 40842** | 75.1ns | 60.2ns |
| Error | 4975 | 134 | 0.549 | 0.0376 | 0.552 | 1.48 | 7893 | 225514 | 9956 | 61.8 | 134 |

**Note.**

ns, *not significant*.

*significant at the 0.05 and 0.01 probability levels, respectively. S.O.V: source of variations, Y: year, R: replication, B: bacteria, M: organic amendments. The Y, B, and M are experimental factors whose main (individual) and interaction effects on the measured traits are included in this table.

**significant at the 0.05 and 0.01 probability levels, respectively. S.O.V: source of variations, Y: year, R: replication, B: bacteria, M: organic amendments. The Y, B, and M are experimental factors whose main (individual) and interaction effects on the measured traits are included in this table.

biomass-C was obtained from VAM under *Azotobacter* inoculation, although it did not statistically different with some treatments (Fig. 3A and 3B).

The number of bacteria was influenced by interaction effect of bacteria × organic amendments ($P < 0.01$) (Table 4). The mean comparison showed that, organic amendments increased the number of bacteria in non-inoculated plants by 22–32%, and it increased the number of bacteria by 22–32% both in plants inoculated with *Azotobacter* and *Azospirillum*. Although, there was no significant difference between VAM and some organic treatments in non-inoculated plants, but in the plants inoculated with *Azotobacter* and *Azospirillum*, the highest number of bacteria was obtained from the VAM (Fig. 4).

## Chlorophyll content

Chlorophyll content of flag leaf was influenced by interaction effect of organic amendments × bacteria (Table 4). The plants treated by N$_2$-fixing bacteria increased the effect of organic amendments on chlorophyll content, so that the content range of chlorophyll *a* was obtained as 4.5–1.9, and 7.3–4.3 mg/L, respectively (Fig. 5) in non-inoculated plants and inoculated plants, and the content range of chlorophyll *b* was obtained as 1.0–0.4, and 0.7–2.8 mg/L, respectively (Fig. 6) in non-inoculated and inoculated plants. Under non-inoculation conditions, although organic amendments of total chlorophylls (chlorophyll *a* + *b*) increased by 16–52% compared to control, but this increase was obtained as 23–79% and 11–67%, respectively (except VM) under *Azotobacter* and *Azospirillum* inoculation conditions. N$_2$-fixing bacteria played a positive role in improving flag leaf chlorophyll content especially in vermicomposts containing *Azolla*. The highest content of total chlorophyll was obtained from VRAM, VRM, and VAM under *Azotobacter* inoculation, and from VRAM under *Azospirillum* inoculation (Fig. 7).
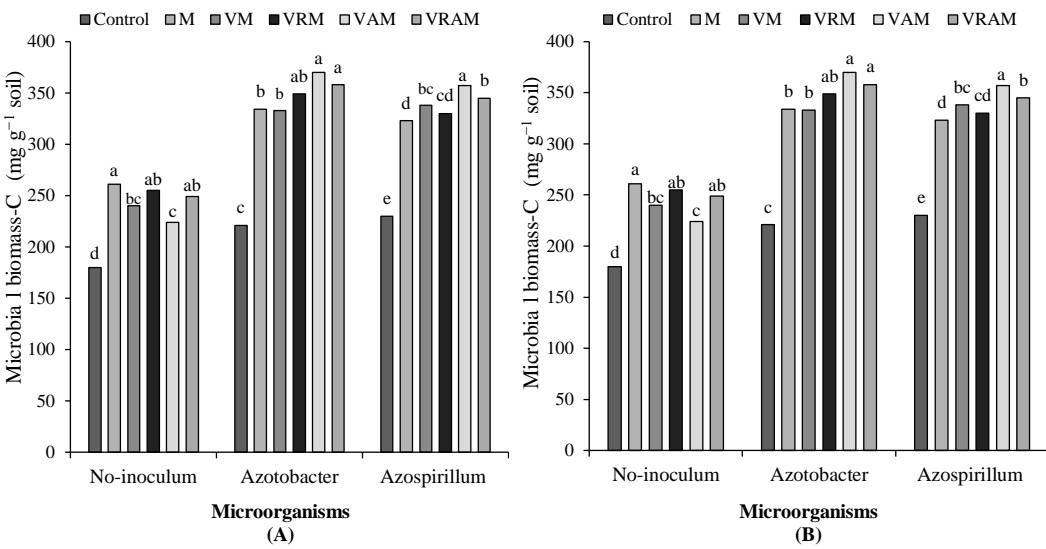

**Figure 3** Mean comparison of effect of organic amendments on soil microbial biomass–C under inoculation with $N_2$-fixing bacteria in 2017 (A) and 2018 (B).

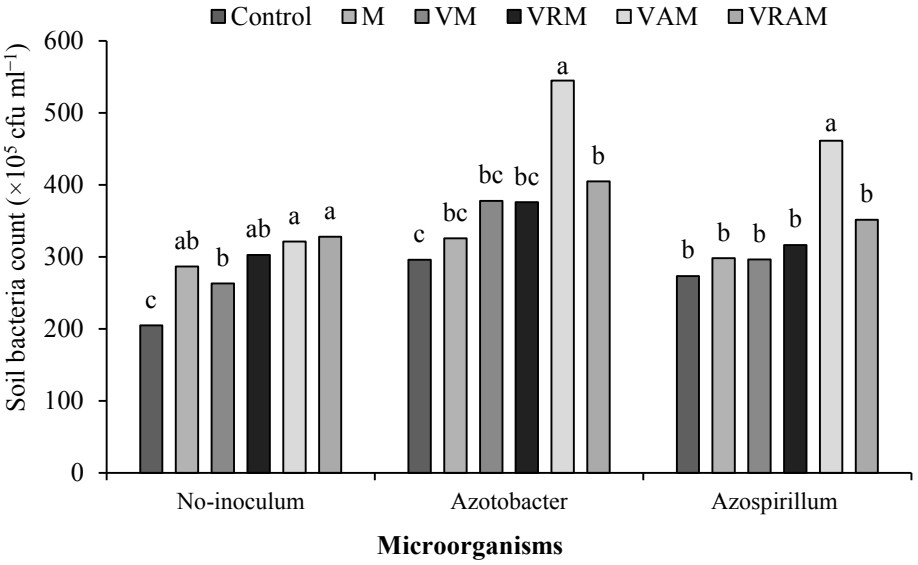

**Figure 4** Mean comparison of effect of organic amendment on viable anaerobic bacteria count ($\times 10^5$ CFU ml) under inoculation with $N_2$-fixing bacteria.

## Tiller number, grain yield, biological yield and HI

Tiller number, grain yield and biological yield were influenced by interaction effect of bacteria × organic amendments × year but HI was influenced by interaction effect of organic amendments × bacteria (Table 4). In both years, organic amendments significantly increased the number of tillers compared to control and in inoculated plants, there were more tillers, so that tillers of organic amendments were obtained as 16–45% under

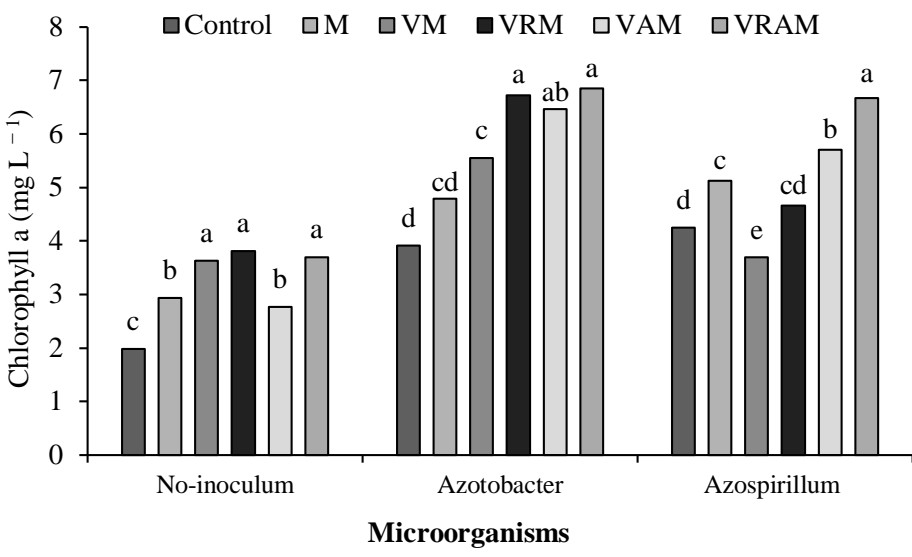

**Figure 5** Mean comparison of effect of organic amendment on chlorophyll a under inoculation with N$_2$-fixing bacteria.

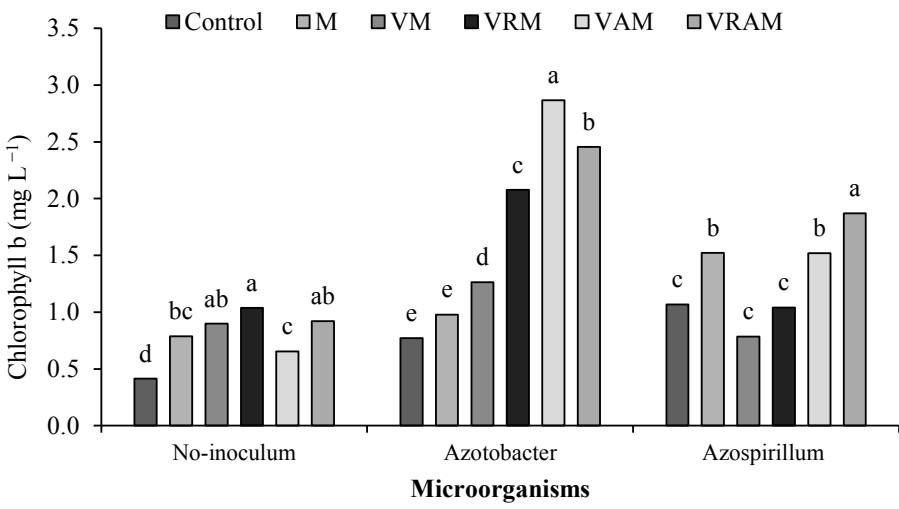

**Figure 6** Mean comparison of effect of organic amendment on chlorophyll b under inoculation with N$_2$-fixing bacteria.

non-inoculation conditions in the first year compared to control and in the second year they were obtained as 19–53%. Under the non-inoculation condition with *Azotobacter* in the first year, they increased by 34–75% and in the second year, they increased by 51–86%. Under the non-inoculation condition with *Azosperilium* in the first year they increased by 29–73% and in the second year, they increased by 19-78%. In both years, the maximum number of tiller was obtained from VAM and VARM under inoculation with *Azotobacter* (Table 5).

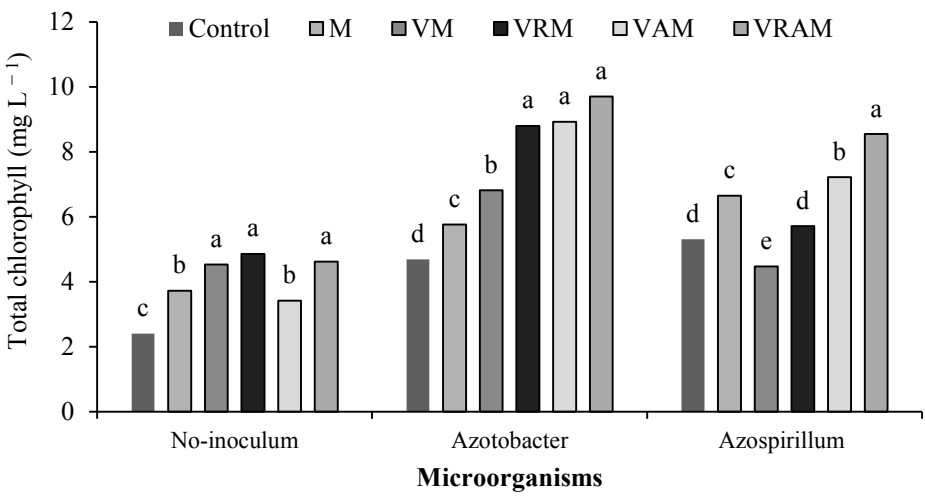

**Figure 7** Mean comparison of effect of organic amendment on total chlorophyll under inoculation with $N_2$-fixing bacteria.

Maximum grain yield was obtained from vermicompost treatments when plants were inoculated with $N_2$-fixing bacteria. Meanwhile, grain yield of control increased by 9.7 and 6.4%, respectively in the first year under inoculation with *Azotobacter* and *Azosporilium*, and in the second year under inoculation with *Azotobacter* and *Azosporilium*, the grain yield increased by 19 and 15%, respectively. But the grain yield increased by 29–56% in the first year and 41–83% in the second year with application of organic amendments under inoculation with *Azotobacter*, and under inoculation with *Azosporilium*, it increased by 20–43% in the first year and by 33–74% in the second year.

It should be noted that, the application of manure without vermicomposting (M) increased grain yield by 6, 29 ,and 20% in the first year ,and 9.4, 41 ,and 33% in the second year, respectively under non-inoculation, inoculation with *Azotobacter* and *Azospirillum* compared to the control, while vermicomposting manure (VM) increased grain yield by 15, 37 ,and 31% in the first year ,and 48, 73 ,and 64% in the second year, respectively under non-inoculation, inoculation with *Azotobacter* and *Azospirillum* compared to the control. In both years, the maximum grain yield was obtained from VAM + *Azotobacter* (Table 5).

The biological yield of the plants changed in response to organic amendments × bacteria, so that the application of organic amendments increased biological yield by 22–58% in the first year and by 28–45% in the second year under non-inoculation conditions, it increased by 44–86% and 31–66%, respectively in the first and second years under the inoculation with *Azotobacter* compared to the control , and increased by 33–50% and 28–52%, respectively in the first and second years under the inoculation with *Azospirillum* compared to the control. Among all organic amendments, VAM (vermicompost of manure + *Azolla*) showed the highest biological yield under both non-inoculation and inoculation conditions (Table 5).

**Table 5** Mean comparison of the effect of organic amendments on plant height, panicle height, no. tillers hill⁻¹, seed yield, biological yield and harvest index under inoculation with N2-fixing bacteria.

| Bacteria | Organic | Plant height (cm) | Panicle height (cm) | No. of tiller hill⁻¹ | Grain yield (kg/hm²) | Protein yield (kg/hm²) | Biological yield (kg/hm²) |
|---|---|---|---|---|---|---|---|
| | | | | **2017** | | | |
| | Control | 110.1e | 25.60e | 12.3i | 2690g | 274h | 5814j |
| | M | 124.3d | 29.66c–e | 15.5hi | 2857fg | 341e | 7095hi |
| | VM | 128.9b–d | 28.27c–e | 18.66c–g | 3095c–g | 332f | 7161hi |
| Non-inoculum | VRM | 126.2cd | 29.62c–e | 18.0d–h | 2881fg | 291gh | 7124hi |
| | VAM | 131.0b–d | 36.56abc | 19.3b–e | 3595b–f | 318f | 8757d |
| | VRAM | 126.3cd | 34.90a–d | 19.0c–f | 3500b–f | 381d | 7490gh |
| | Control | 124.2d | 29.63c–e | 16.3h | 2985d–g | 278gh | 7833e–h |
| | M | 131.7b–d | 30.63c–e | 16.66gh | 3476b–g | 331f | 8381d–f |
| Azetobacter | VM | 134.8a–c | 33.00b–e | 20ab–d | 3690b–e | 334ef | 9890bc |
| | VRM | 132.0b–d | 28.85c–e | 18.0d–h | 3428b–g | 338e | 9047cd |
| | VAM | 143.5a | 43.22a | 22.6a | 4667a | 438a | 10833a |
| | VRAM | 137.9ab | 40.37ab | 21.6ab | 4214ab | 366d | 10057ab |
| | Control | 114.1e | 27.9de | 14.7hi | 2905efg | 293g | 7528fgh |
| Azospirillum | M | 126.7cd | 29.52c–e | 17.0f–h | 3238c–g | 333ef | 8347d–g |
| | VM | 130.3b–d | 34.01b–d | 17.667e–h | 3547b–f | 347e | 7771e–h |
| | VRM | 129.4b–d | 35.61a–d | 18.0d–h | 3295c–g | 371d | 6505ij |
| | VAM | 135.6a–c | 39.06ab | 21.3ab | 3857bc | 424b | 9209b–d |
| | VRAM | 133.3b–d | 39.83ab | 20.6a–c | 3714b–d | 418bc | 8524de |
| | | | | **2018** | | | |
| | Control | 117.6i | 27.10d | 11.6h | 2309j | 249i | 6309i |
| | M | 126.0f–h | 28.58cd | 16.33fg | 2928g–j | 190j | 8095gh |
| | VM | 126.7f–h | 35.13a–d | 18def | 3428e–h | 242i | 8566d–g |
| Non-inoculum | VRM | 126.2f–h | 30.41cd | 17.01ef | 3214f–i | 283h | 8467e–h |
| | VAM | 126.0f–h | 33.87a–d | 18.0d–f | 3942b–d | 340d–f | 9200b–d |
| | VRAM | 131.3d–g | 40.13ab | 17.33ef | 3666c–f | 280h | 8809d–f |
| | Control | 124.5g–i | 27.26d | 13.5gh | 2761h–j | 320fg | 7847h |
| | M | 131.5d–g | 29.45cd | 18def | 3523d–g | 351c–e | 8309f–h |
| Azetobacter | VM | 132.0d–f | 31.54b–d | 19.33b–e | 4100b–e | 296gh | 9680a–c |
| | VRM | 131.7d–f | 28.82cd | 21.33a–c | 3976b–e | 332ef | 8462e–h |
| | VAM | 150.2a | 40.44a | 22.33a | 5081a | 404a | 10323a |
| | VRAM | 142.2bc | 36.74a–c | 21.33a–c | 4452ab | 352c–e | 9753ab |
| | Control | 120.1hi | 27.85d | 12.5h | 2667ij | 290h | 6095i |
| Azospirillum | M | 127.6e–g | 29.26cd | 17.0ef | 3500d–g | 303gh | 8123f–h |
| | VM | 130.3d–g | 34.45a–d | 18.66c–f | 3857b–f | 367bc | 8795d –f |
| | VRM | 134.1de | 34.68a–d | 18def | 3782b –f | 364b–d | 8605d–g |
| | VAM | 135.6cd | 36.59a–c | 21.66ab | 4357bc | 379b | 9623bc |
| | VRAM | 143.3ab | 36.61a–c | 20.66a–d | 4405ab | 380ab | 9033c–e |

## Protein yield

The interaction effect of organic amendments × bacteria × year on protein yield was significant ($P < 0.01$) (Table 4). In both years, organic amendments had significant

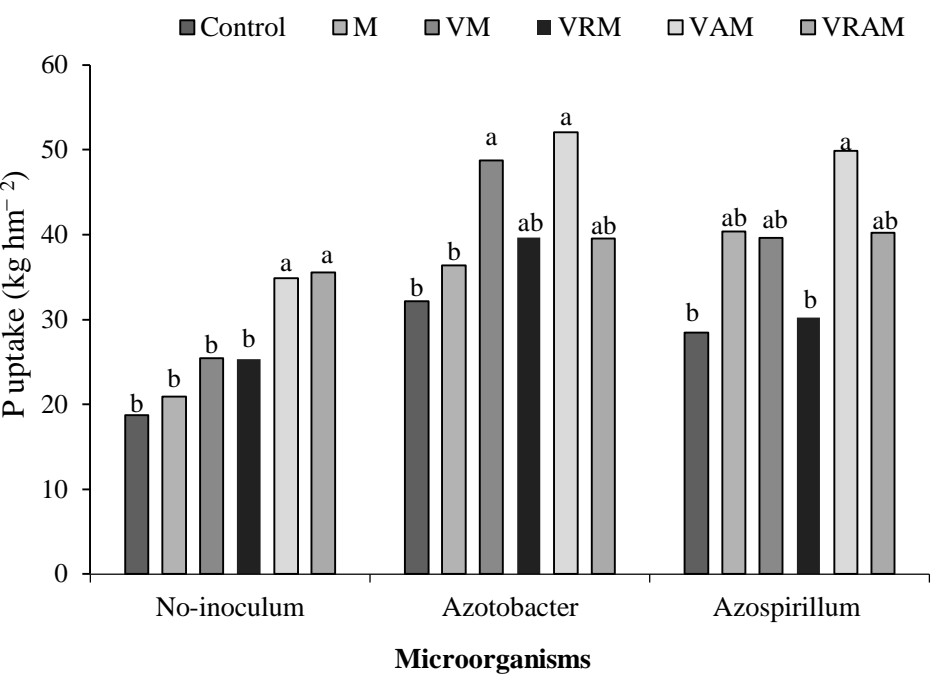

**Figure 8** **Mean comparison of the effect of organic amendment on P uptake under inoculation with N$_2$-fixing bacteria.**

superiority compared to the control. In both years, under non-inoculation and inoculation with *Azotobacter* the highest protein yield was obtained from plants treated with VAM, but under inoculation conditions with *Azospirillum*, there was no significant difference between VAM and VRAM. Among all treatments, in both years, the highest protein yield was obtained from *Azotobacrer* + VAM (438 kg/hm$^2$ in the first year and 404 kg/hm$^2$ in the second year). It is worth noting that, although grain yield of treatments was less in the first year compared to the second year, but the protein yield was higher in the first year (Table 5).

## P and K uptake

K and P uptake by grain were influenced by interaction effect of organic amendments × bacteria (Table 4). The mean comparison of interaction effect of organic amendments × bacteria on P uptake showed highest P uptake was obtained under non-inoculation conditions by VAM and VARM (with a 76 and 79% increase compared to the control), under inoculation conditions with *Azotobacter* by VAM and VM (with a 44 and 35% increase compared to the control + *Azotobacter*), and also the under inoculation conditions with *Azospirillum* by VAM (with a 63% increase compared to the control + *Azospirillum*) (Fig. 8).

The mean comparison of the interaction effect of organic amendments × bacteria on K uptake showed highest K uptake under non-inoculation conditions, and under inoculation with *Azospirillum* by VAM (with a 51% increase compared to control), and VRAM and VAM (with a 80 and 74% increase compared to control + *Azospirillum*). But in plants

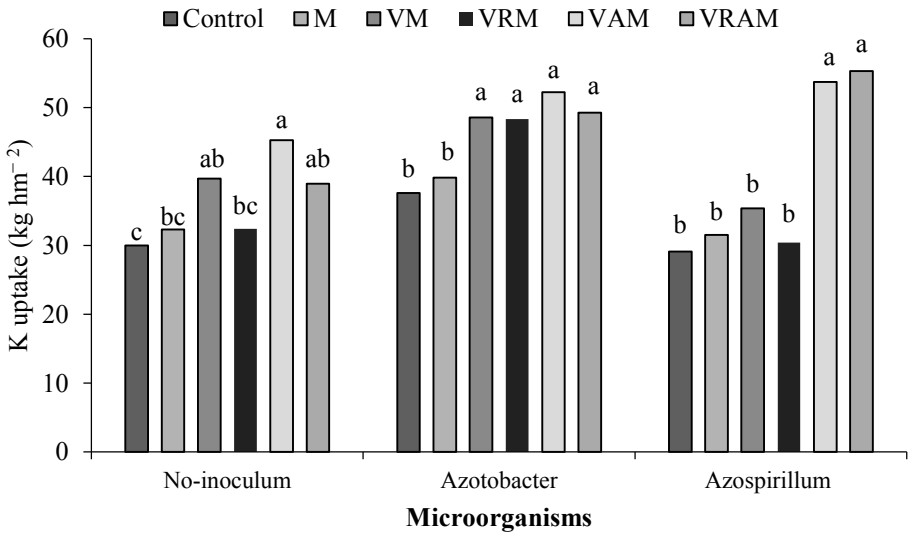

**Figure 9** Mean comparison of the effect of organic amendment on K uptake under inoculation with N$_2$-fixing bacteria.

inoculated *with Azotobacter*, there was no significant difference between vermicompost treatments including VM, VM, VAM, and VARM, but significantly increased K uptake compared to the manure treatment (M) and control (Fig. 9). Also, control + *Azotobacter* increased K and P uptake by 25 and 32%, respectively, and control + *Azosperilum* increased K and P uptake by 8.8 and 19%, respectively compared to control + non-inoculation.

## DISCUSSION

The results showed that, vermicomposting of manure with and without *Azolla* and rice straw and their application as organic fertilizers under inoculation with N$_2$-fixing bacteria, especially *Azotobacter*, and increased the number and biomass-C of soil microorganisms and urease enzyme activity. The number and biomass-C of soil microorganisms and urease enzyme activity were lowest in control plots due to high stress and inadequate nutrient supply, lack of using organic fertilizer and lower amounts of rhizodeposition (root exudates and root biomass). These results are consistent with the findings of the studies by *Cooper & Warman (1997)*; *Bhattacharyya, Chakrabarti & Chakraborty (2005)*; *Kopecky et al. (2011)*; and *Tu et al. (2017)*. An increase in the number and biomass-C of microorganisms in planted rice under flooded condition has been reported following the application of vermicompost (*Nayak, Babu & Adhya, 2007*). *Albiach et al. (2000)* and *Cui et al. (2018)* reported that, annual application of adequate amounts of some organic residues led to significant increase in soil enzyme activities.

The inoculation of plants with *Azotobacter* and *Azospirillum* increased the application efficiency of organic fertilizers in improving the number and biomass-C of soil microorganisms and urease enzyme activity. The positive role of N$_2$-fixing bacteria in increasing the population of Diazotrophic bacteria, improving enzyme activity and microbial biomass N in rice rhizosphere has been confirmed in the studies by *Kumar,*

*Swain & Bhadoria (2018)* and *Zhang et al. (2018)*. *Mahanta et al. (2012)* reported that, vermicomposting of manure increased the soil microbial activity by 79% and biomass-C by 68% compared to manure, resulting from improvements in soil physical and chemical activity by vermicompost.

In both years of experiment and in all treatments, the urease activity increased to stage T3 ($\frac{1}{4}$ panicle initiation stage), due to increased microbial activity of the soil, but decreased in the stage T4 ($\frac{1}{4}$ maturity). Although, the organic treatments increased the urease activity, but inoculation of plants with *Azotobacter* and *Azospirillum* increased the effect of organic amendments on urease activity and among the treatments, VAM + *Azotobacter* showed the most urease activity. In agreement with our results, *Rodrigues, Ladeira & Arrobas (2018)* reported that, the organic material containing *Azotobacter* increased the bioavailability of N over organ, by an additional $N_2$-fixing value of 11.4 kg/hm$^2$ estimated from the six crops of the field experiment. *Nayak, Babu & Adhya (2007)* reported increased urease activity with a compost made of residues of rice + urea in flooded rice, and *Arora & Kaur (2019)* reported about *Aspergillus terreus*-enriched organic manures. The increase of total organic carbon, water-soluble carbohydrate, water-soluble C, and proliferation and microbial activity of the soil are among the most important contributions of organic amendments (*Srivastava, Aragno & Sharma, 2010*) resulting in the release of elements in the soil and organic materials and making available roots by increasing the bacterial activity (*Pérez-Piqueres et al., 2006*). In fact, changes in the composition of microbial communities have been observed as a result of incorporating inorganic or organic amendments (*Marschner, Kandeler & Marschner, 2003*; *Crecchio et al., 2004*). But various properties of organic materials can have a different effect on soil microbiota and will strongly influence the microbial use of the C contained in these materials. Vermicomposting increased the number and biomass-C of microorganisms of the soil, especially under inoculation with *Azotobacter*, compared to the manure treatment (M) and control. Carbohydrates would act as sources of energy in respiration of microbiota (*Zhong et al., 2010*; *Li et al., 2015*) showing intense temporal changes due to their continuous process of synthesis and degradation. This is reflected by the changes in urease activity during the growing season of rice under various treatments. During the growing season, urease activity was higher in treatments under inoculation compared to non-inoculation treatments, especially VAM and VRAM treatments that showed more enzyme activity under inoculation with *Azotobacter* compared to *Azospirillum*. Higher urease activity under the application of VAM and VRAM treatments demonstrated higher N -fixing ability of *Azotobacter*, which increased the available N content in the rice rhizosphere.

*Zhang et al. (2017)* and *Islam et al. (2012)* also suggested that, wide variety of free-living $N_2$-fixing bacteria can be used as a feasible alternative to N fertilizer in rice ecosystems. The addition of organic material to soil results in an increase in the total organic carbon content increasing the proliferation and activity of microorganisms (*Martinez-Balmori et al., 2013*), therefore in the beginning of the growth, although the enzyme activity of microorganisms is high, but they take out the nutrients from the plant and spend them for their bodies (*Rodrigues, Ladeira & Arrobas, 2018*).

Therefore, the reason for the superiority of *Azolla*-containing organic treatments, namely VAM (vermicompost of manure + *Azolla*) and VRAM (vermicompost of manure + *Azolla* + rice straw), can be due to more amounts of N compared to other treatments. The application of *Azolla* compost in paddy rice has been studied in many countries and its positive effect in increasing the crop has been well proved (*Razavipour et al., 2018*). The preferred effect of VAM and VRAM on the microbial population of the soil and the urease activity, as well as the growth and yield of rice in inoculated plants is evident. In this regard, *Zhang et al. (2018)* reported that, inoculation of *A. brasilense* and *P. fluorescens* in the rice rhizosphere accelerated N transformations and improved the N-supplying capacity of the rhizosphere soil, and increased rice biomass. The most beneficial effects were observed with *A. brasilense* and *P. fluorescens* co-inoculation in the rice rhizosphere. Inoculation of microorganisms is influenced by agricultural practices (*Velusamy, Immanuel & Gnanamanickam, 2013*) improving both growth and nutrient acquisition (*Khan, 2018*) of rice grown under upland conditions (*Rajeshkannan, Sumathi & Manian, 2009*; *Zhang et al., 2017*).

The used treatments have improved the biological, physical, and chemical properties of the soil and increased the growth of morphological traits. So that in both years, under both non-inoculation and inoculation conditions, application of organic amendments compared to control increased plant and panicle height. The highest plant and panicle height was obtained from plants treated with VAM and VRAM under inoculation with *Azotobacter*. *Kumar, Swain & Bhadoria (2018)* and *Lin, Zhu & Lin (2011)* reported that, the number of functional leaves, leaf area and total number of tillers was higher in plants treated with organic nutrient, which increased the photosynthetic rate leading to higher plant height.

Moreover, the increase in photosynthesis of the plant depends on the content of leaf chlorophyll (*Murchie et al., 2002*; *Singh, Singh & Sharma, 2013*) as influenced by organic amendments and bacterial inoculation in this experiment, and the highest chlorophyll of *a*, *b* and total chlorophyll was observed in vermicompost treatments under inoculation with *Azotobacter*. In most studies, the increase in chlorophyll content of leaf has been reported to be directly associated with the absorption of N and Fe (*Hosseinzadeh, Amiri & Ismaili, 2018*). In addition to atmospheric N fixation, the solubility of minerals such as P, K, Cu, and Fe and the production of sidrophor are among the important effects of *Azotobacter* and *Azospirillum* (*Yadav et al., 2014*; *Rodrigues, Ladeira & Arrobas, 2018*). On the other hand, vermicompost is also rich in some macro and micro elements (*García et al., 2014*; *Yadav et al., 2014*; *Eo & Park, 2019*). Therefore, vermicompost involves desirable properties such as a high capacity for cation exchange, increased absorption of nutrients and other beneficial physical, biological, and chemical properties (*Ludibeth, Marina & Vicenta, 2012*) increasing the chlorophyll content and stability of the photosynthetic system of rice plants .The vermicompost application led to a decrease in the Reactive Oxygen Species (ROS) production, increased accessibility to nutrients and the required elements for biochemical activity (*Baghbani-Arani & Modarres-Sanavy, 2017*). *Mahanta et al. (2012)* and *Shirkhani & Nasrolahzadeh (2016)* reported increased chlorophyll content of leaves by organic fertilizers and $N_2$-fixing bacteria. Increasing the leaf chlorophyll content

leads to an increase in photosynthetic capacity. The acceleration of photosynthesis leads to an increase in plant height, number of tillers and number of grain per panicle (*Yi-hu et al., 2014*). *García et al. (2016)* reported that, vermicompost increased the number of tillers and plant dry weight by producing humic acid and supplying nutrients, especially N and K.

*Meena & Shivay (2010)* introduced the improvement of rhizosphere in increasing the root activity as a factor increasing the number of tillers. In our experiment, in both years, bacterial treatments × organic amendments had a positive effect on the number of tillers. Although the number of tillers is dependent on environmental and genetic factors, but environmental conditions, especially nutrition have highest effect in the early stages of growth (*Murchie et al., 2002*; *Nuemsi et al., 2018*). In both years, vermicomposts containing *Azolla* under inoculation with *Azotobacter* produced greatest tillers. Previous studies revealed that, these management practices have amended the soil structure and function, as well as nutrient availability influencing the plant height, number of tiller, panicle length, biological and grain yield of rice by the synergistic effects (*Simarmata et al., 2016*; *Tsujimoto et al., 2009*; *Thakur et al., 2010*; *Kumar, Swain & Bhadoria, 2018*). The combined use of vermicompost + $N_2$-fixing bacteria is better than sole application of them for grain, straw and biological yields, because organic fertilizers can reduce N loss (*Kumar & Singh, 2001*) and maintain the supply of N to rice plants for a longer time (*Nayak, Babu & Adhya, 2007*; *Li et al., 2015*).

In the present experiment, in both years, the grain and biological yield of inoculated plants increased significantly by applying different vermicomposts compared to non-inoculated plants under similar treatments. The highest grain yield by VAM + *Azotobacter* treatment in the first and second years was obtained as 4,667 and 5,081 kg/hm$^2$, respectively, although there was no statistically significant difference in some of the treatments. In both inoculated and non-inoculated plants, VAM treatment produced the highest biological yield and VRAM treatment ranked the next, in addition, the maximum biological yield was observed under inoculation with *Azotobacter*. *Eo & Park (2019)* also stated the role of vermicompost in increasing rice growth and yield, so that the vermicompost application has altered the resource allocation and soil chemical properties leading to establishment of new interactions between root parameters and components. In most studies similar to our study, the main cause of increase in the growth and grain yield of rice by applying organic fertilizers under inoculation with $N_2$-fixing bacteria has been introduced due to the increase in the availability and supply of N in different stages of rice growth. In our experiment, due to the application of bacteria × vermicompost, the amount of fixed atmospheric $N_2$ and organic materials content was enhanced in the soil.

Marginal increase in N content of rice straw due to inoculation *with Azotobacter* and use of organic fertilizers has been reported (*Rodrigues, Ladeira & Arrobas, 2018*), resulting from an increase in the N availability through synchronized released from the inoculation of $N_2$-fixing bacteria, which increased the N concentration proportionately in grains and straw and finally led to higher N uptake with the highest amount of N (*Li et al., 2015*; *Tu et al., 2017*). In our study, in both years, protein yield significantly increased in plots treated with both vermicompost under inoculation and non- inoculation conditions compared to control and M treatments. In non-inoculated and inoculated plants with *Azotobacter*,

the highest protein yield was obtained from VAM, but there was no significant difference between VAM and VRAM in plants inoculated with *Azospirillum*. *Tejada & González (2009)*, *Bejbaruah, Sharma & Banik (2013)*, *Mengi et al. (2016)* and *Taheri Rahimabadi, Ansari & Razavi Nematollahi (2018)* have reported the effect of vermicompost and organic amendments on increasing the grain protein of rice. The higher rice yield using the integrated N management might be due to the increase in number of filled grains per panicle, the number of panicles per plant, and 1,000-grain weight (*Amanullah, 2016*). The difference in nutrient uptake from different N source combinations significantly influences growth and yield potential (*Gu et al., 2014*). *Myint et al. (2010)* reported that, organic amendments can increase yield through the improvement of soil water holding capacity, physical and chemical conditions, reducing volatilization of nitrogenous fertilizers to $NH_3$ gas and the greater availability of plant nutrients for a longer time.

The P and K uptake of grain were influenced by interaction effect of organic amendments × bacteria. The highest P uptake in non-inoculated plants was obtained from VAM and VARM (with a 76 and 79% increase compared to control). In plants inoculated with *Azotobacter*, VAM and VM treatments showed the highest P uptake, but in plants inoculated with *Azospirillum*, VAM treatment showed the highest P uptake. The highest K uptake in non-inoculated plants was obtained from VAM treatment and it was obtained in plants inoculated with *Azospirillum* from VRAM and VAM. But in plants inoculated with *Azotobacter*, there was no significant difference between vermicompost treatments including VM, VRM, VAM, and VARM, but significantly increased K uptake compared to the manure (M) and control. Our results confirm the findings of the study by *Yadav et al. (2014)* who have reported 50% increase in content and uptake of N, P, and K in wheat using PGPR strains. An increase in P uptake over control may be due to the bacterial solubilization of insoluble phosphate in soil. These bacteria showed an effective role in P uptake and growth promotion of plants by dissolution of inorganic insoluble phosphate as reported in the study by *Narula et al. (2000)*.

These results showed that, the application of organic amendments + growth stimulating bacteria leads to an increase in organic material pools and nutrient availability, as well as amendment of the physical environment of soil and rice yield (Chen 2017). Additionally, the application of vermicompost, especially VAM and VARM treatments, either alone or together with *Azotobacter* and *Azospirillum*, in addition to increasing the number and biomass-C of soil microorganisms, leads to improvement of the solubility of nutrients, especially N, P, K and some microelements.

In line with our results, *Zhong et al. (2010)*, *Li et al. (2015)*; *Wang et al. (2015)* and *Cui et al. (2018)* reported that, the application of organic fertilizers influences on soil physicochemical and biological properties, especially regarding soil pH and microbial biomass C. *Nayak, Babu & Adhya (2007)* showed that, soil urease activity had a significant negative relationship with Eh and had a positive relationship with the content of N, P, and Fe. Therefore, application of vermicompost and $N_2$-fixing bacteria by changing the state of redox and soil pH and catalytic efficiency may influence the accumulation and activity of enzymes, and importantly increases the availability of nutrients (*Wlodarczyk, Stepniewski & Brzezinska, 2002*). In this regard, *Basha, Basavarajappa & Hebsur (2017)*

and *Taheri Rahimabadi, Ansari & Razavi Nematollahi (2018)* reported the effect of vermicompost application on increasing the biomass and grain yield of rice due to the increased absorption of essential elements of plant growth. Similarly, *Yadav et al. (2014)* reported that, Azotobacter and *Azospirillum* inoculated with rice seedling led to a 35–21% increase in the yield and yield components, respectively compared to control under field condition. There were significant interactions between nitrogen level and biofertilizer regarding yield, yield component, and protein content (*Nosheen et al., 2016*). In this study, *Azotobacter* and *Azospirillum* also acted complementarily regarding positive effects of vermicomposts on nutrients absorption and rice yield.

## CONCLUSION

The results of the present study showed that, vermicomposting of cow manure, rice straw and *Azolla* led to improvement of physicochemical properties of used materials. In both years, the application of vermicomposts had a positive effect on the number and biomass-C of soil microorganisms and urease enzyme activity. It was found that, these changes led to the amendment of physical and biological properties of the soil and increased the chlorophyll content, biological and grain yield, and the uptake of N, P, and K by the availability of nutrients. In this study, vermicomposts containing *Azolla* (VAM and VARM) were more beneficial for increasing the rice productivity. Inoculation of the plants with *Azotobacter* and *Azospirillum* also showed significant superiority in the above mentioned traits compared to non-inoculated plants. On the other hand, the synergistic effect of bacteria with organic amendments, especially with *Azotobacter* was observed with vermicomposts containing *Azolla*, so that the highest grain yield in the first and second years with an average of 4667 and 5081 kg/hm$^2$, respectively was obtained from VAM + *Azotobacter*. The present study provides information on the effect of applying manure and various vermicomposts on the number and biomass-C of soil microorganisms and urease enzyme activity and finally on the absorption of elements and grain yield under inoculation with $N_2$-fixing. Our results suggested that, applying organic source for vermicomposting has a great effect on the enzymatic and biological activity of the soil under flooded conditions contributing to the sustainable development of agro-ecosystems. It is suggested to evaluate the vermicomposting effect on various organic sources weak and rich in nutrients, especially N, along with inoculation of plants or vermicompost enrichment with $N_2$-fixing bacteria in future studies.

### Funding
This research was partially supported by a research grant (UOZ-GR-9618-25) provided by University of Zabol. The funders had no role in study design, data collection and analysis, decision to publish, or preparation of the manuscript.

### Grant Disclosures
The following grant information was disclosed by the authors:

Zabol University: UOZ-GR-9618-25.

## Competing Interests

The authors declare there are no competing interests.

## Author Contributions

- Mehdi Ghadimi conceived and designed the experiments, performed the experiments, prepared figures and/or tables, authored or reviewed drafts of the paper, and approved the final draft.
- Alireza Sirousmehr and Ahmad Ghanbari conceived and designed the experiments, authored or reviewed drafts of the paper, and approved the final draft.
- Mohammad Hossein Ansari conceived and designed the experiments, analyzed the data, prepared figures and/or tables, authored or reviewed drafts of the paper, and approved the final draft.

## Data Availability

Raw data are available in a Supplemental File.

## Supplemental Information

Supplemental information for this article can be found online at http://dx.doi.org/10.7717/peerj.10833#supplemental-information.

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
