# Peer review of "Organic soil amendments using vermicomposts under inoculation of N2-fixing bacteria for sustainable rice production"

_PeerJ, doi:10.7717/peerj.10833_

## Round 0.1 · original submission · Major Revisions

Dear Dr. Ghadimi and colleagues:

Thanks for submitting your manuscript to PeerJ. I have now received two independent reviews of your work, and as you will see, the reviewers raised some concerns about the research. Despite this, these reviewers are optimistic about your work and the potential impact it will have on research studying vermicomposting and microbial applications for rice production. Thus, I encourage you to revise your manuscript, accordingly, taking into account all of the concerns raised by both reviewers.

While the concerns of the reviewers are relatively minor, this is a major revision to ensure that the original reviewers have a chance to evaluate your responses to their concerns.

There are many comments by both reviewers that ask for more information on specific issues; please address these. Please consider moving some of the material to the supplements.

I look forward to seeing your revision, and thanks again for submitting your work to PeerJ.

Good luck with your revision,

-joe

·

Basic reporting

Clearly presented

Experimental design

A robust design, with strong data

Validity of the findings

These are very clear and useful. The treatment effects were well described and trends effectively defined.

Additional comments

A very good paper and worth publishing. I have some minor comments as follows:

Line 59 - suggest '...and grain yield has a...'
Line 87 - '...spiked wheat...' please clarify the term spiked for readers
Line 133 - '...and 15 m' Is this altitude?
lines 154-160. This paragraph is rather confusing - please re-write more clearly
Line 164 - suggest '... reactor, which supported a population density of 250 g of earthworms...'
Line 165 - suggest '...reactor was divided...'
line 167 - suggest '...which was processed...'
Lines 201-2 This sentence is confusing - please re-write
line 216 - '...soil was paved...' is 'taken' a better word than 'paved'? '..at the maximum stage...' - what stage?
Line 219 - (Witt et al, 2000).
Line 225 - suggest 'Kelly et al,1999
Line 248 - '...dough stage...' - please clarify
Line 250 - '...the claw...' - please clarify

Results - please clarify that treatment differences stated in the text are statistically valid and not conjectural
Height of panicle and plant section - lines 332-338. Table 4 is referred to - is that meant to be Table 3?
Line 376 - at end of sentence suggest refer to table 4.
Line 390 - Azospicilum - check spelling
Line 440 - suggest 'In agreement with...'
Line 526 - suggest '...for a longer...'
Line 568 - suggest '...treatment and it was...'
Line 620 - suggest '...sources weak and...'

Table 2 - please specify +- values - what error values are they?
Table 3 - Left column treatment abbreviations need a key to clarify
Figures - please check and clarify error bars throughout
Figure 3 - I found this difficult to read and interpret

References - the following references are cited in the text but missing from the Reference list:
Line 100 - Sanati et al, 2011
Line 267 - AOAC 1970

Reviewer 2 ·

Basic reporting

Organic fertilizers provide essential nutrients to plant. This study reports the effect of vermicomposting of cattle manure mixture with Azolla and rice straw on soil microbial activity, nutrient uptake, and grain yield under inoculation of N2−fixing bacteria using a field experiment in a paddy soil. The manuscript is of interest and generally well written. The data are enough to support their conclusions. But there are some unclear points that should be addressed prior to considering it for publications. Overall, this manuscript is recommended for Peer J after Minor Revision.

Please find some more detailed comments below.

1. Lines 167-170, in general, the optimum moisture content of vermicomposting is about 60%. Why the authors chose 75-80% of moisture content?
2. Lines 171-178, how about the temperature development during vermicomposting? Is the temperature suitable for the survival of earthworms? Also, is the temperature enough to kill pathogens in manures?
3. Figure 3, it is better to use the specific days after transplantation rather than rice growth stages.
4. The total number of tables and figures are 16. It is suggested to move some data into Supporting Information.

Experimental design

no comment

Validity of the findings

no comment

Additional comments

no comment

---

## Round 0.2 · accepted · Accept

Dear Dr. Ghadimi and colleagues:

Thanks for revising your manuscript based on the concerns that were raised. I now believe that your manuscript is suitable for publication. Congratulations! I look forward to seeing this work in print, and I anticipate it being an important resource for groups studying vermicomposting and microbial applications for rice production. Thanks again for choosing PeerJ to publish such important work.

Best,

-joe

·

Basic reporting

No comment

Experimental design

No comment

Validity of the findings

No comment

Additional comments

The amendments to the m/s appear to meet the original comments made. I am happy to recommend publication.